

# Effect of isolation on coat colour polymorphism of Polynesian rats in Island Southeast Asia and the Pacific

Alexandra A.E. van der Geer

Naturalis Biodiversity Center, Leiden, Netherlands

## ABSTRACT

Populations of vertebrate species introduced onto islands regularly develop similar phenotypic changes, e.g., larger or smaller body size, shortened limbs, duller coats, as well as behavioural changes such as increased tameness and reduced flight-initiation distance. These changes overlap in part with those associated with the 'domestication syndrome', especially tameness and changes in coat patterns, and might indicate a similar neural crest involvement in the concurrent development of multiple phenotypic traits. Here I examine long-term data on free-living populations of wild Polynesian rats from seven mainland countries and 117 islands ($n = 3{,}034$), covering the species' native and introduced range. Mainland populations showed no aberrant coat patterns, with the exception of one albino, whereas aberrant coat patterns were found in 12 island populations. Observed coat colour polymorphisms consisted of leucistic (including singular white patches), melanistic (darkly pigmented) and piebald (mixed) coat patterns. After isolation for at least seven centuries, wild Polynesian rat populations on islands seem to exhibit a trend towards a higher incidence of aberrant coat patterns. These phenotypic changes are here explained as a neutral, non-adaptive process, likely part of the 'domestication syndrome' (via the commensal pathway of domestication), in combination with genetic drift, little or no gene flow between the islands and/or the mainland and a relaxed selection (as a result of the weakening or removal of competitor/predator pressure) under commensality.

## INTRODUCTION

Evolution on islands has often led to increased phenotypic variation and phenotypic novelty not observed in closely related mainland forms, inspiring studies on evolution ever since Alfred Wallace wrote his influential book (*Wallace, 1880*). The perhaps most obvious phenotypic variation is observed in body mass, especially in island endemic lineages of the Pleistocene epoch, when rabbit-sized rats and pony-sized elephants and hippos were common elements of island faunas (*Van der Geer et al., 2010*). Body mass evolution of island mammals typically follows a graded trend from gigantism in small-sized species to dwarfism in large-sized species (*Foster, 1964*; *Van Valen, 1973*; *Lomolino, 1985*; *Lomolino, 2005*), where the degree and direction of body size evolution vary in a

Corresponding author
Alexandra A.E. van der Geer,
alexandra.vandergeer@naturalis.nl

predictable manner in accordance with characteristics of the ancestral taxon and the levels and nature of various factors like competition, predation, ecological complexity, climate, latitude, island area and isolation (*Lomolino et al., 2012*; *Lomolino et al., 2013*). Other morphological features regularly shared by phylogenetically distant fossil island mammals are proportionally shortened limbs (*Sondaar, 1977*), reduced brain size (*Weston & Lister, 2009*; see, however, *Lyras et al., 2006*), increased convolution of the brain cortex (*Desachaux, 1961*; *Falk et al., 2005*), fusions of limb bones (*Leinders & Sondaar, 1974*; *Moyá-Solá, 1979*; *Van den Bergh et al., 2008*; *Van der Geer, 2014*), high-crowned teeth (*Bover, 2004*; *Van den Bergh, 1999*; *Van der Geer et al., 2006*; *Van der Made, 1999*), loss of dental elements (*Leinders, 1984*; *Lyras et al., 2006*; *Boekschoten & Sondaar, 1972*), shortened muzzles (*Van der Geer, 2005*; *Quintana, Köhler & Moyà-Solà, 2011*; *Kaifu et al., 2011*; *Van der Geer et al., 2018*), increased stereoscopy (*Van der Geer, 2005*) and increased size variation (*De Vos & Van der Geer, 2002*; *Van der Geer, 2014*). Isolated mammal populations may also show shared ecological differences such as increased tameness or ecological naiveté (see also below), higher and more stable population densities, reduced intraspecific aggression, reduced reproductive output (see, however, *Raia, Barbera & Conte, 2003*), who suggested increased reproductive productivity in a dwarf elephant) and reduced dispersal abilities. The suite of ecological and morphological characters often associated with isolated mammal populations is referred to as the 'island syndrome' (*Adler & Levins, 1994*). Other groups, e.g., lizards, also show trait shifts that are consistent with the 'island syndrome', such as a higher degree of melanism (darkly pigmented skins) (*Runemark et al., 2010*), increased tameness (*Cooper Jr & Pérez-Mellado, 2012*), shorter flight initiation distance and reduced sprint speed (*Vervust, Grab & Van Damme, 2007*), smaller clutch sizes (*Huang, 2007*) and larger bodies (*Vervust, Grab & Van Damme, 2007*; *Pafilis et al., 2009*). In parallel, insular birds may have smaller clutch sizes (*Cody, 1971*), dull colours (*Omland, 1997*), increased tameness (*Blondel, 2000*), or may become flightless (*Roff, 1994*).

The concomitance of a set of phenotypic alterations, often similar in unrelated groups, is also a common theme in domestication, already noted by *Darwin (1868)*. In mammals, this linked set of alterations has been called the 'domestication syndrome' (*Wilkins, Wrangham & Fitch, 2014*), and includes increased tameness, shortening of the rostrum, reduced brain size relative to body size (*Kruska, 2005*), floppiness of the ears, curliness of the tail and depigmented skin and fur (*Herre & Röhrs, 1990*). Breeding experiments on silver foxes, *Vulpes vulpes* (*Linnaeus, 1758*) since 1959 (*Belyaev, 1969*) showed that selection for tameability resulted not only in the desired increased tameness, but also in aberrant pigmentation, floppy ears, rolled and shortened tails, shorter limbs, shortened, flattened and widened rostra, smaller brains, earlier onset of sexual maturity and the ability to perceive human gestures, indicating that these phenotypic features are coupled (*Trut, 1999*; *Trut, Plyusnina & Oskina, 2004*; *Trut, Oskina & Kharlamova, 2009*; *Hare et al., 2005*). Similar experiments on brown rats, *Rattus norvegicus* (*Berkenhout, 1769*), initiated by Belyaev as well also yielded tame and aggressive strains, where, as in the foxes, tame rats have shorter, flatter and wider muzzles (*Singh et al., 2017*) and a higher frequency of white spots than the aggressive rats (*Albert et al., 2009*).

This morphological covariation of phenotypic features in domestic animals is generally explained as due to particular changes of the neural crest cell migration (*Crockford, 2002*; *Montague et al., 2014*; *Wilkins, Wrangham & Fitch, 2014*; *Sánchez-Villagra, Geiger & Schneider, 2016*). The selection for tameability results in developmental deficits in the neural crest cells (multipotent stem cells that arise from the dorsal part of the neural tube in vertebrate embryos), which cause most of the characteristics of the 'domestication syndrome' (*Wilkins, Wrangham & Fitch, 2014*). The variation in pigmentation in domesticates is then likely a pleiotropic effect of alleles influencing traits that are related to tameability.

In this respect it is interesting to note the similarity regarding 'naiveté' between the 'island syndrome' and the 'domestication syndrome'. Native island species are often ecologically naive relative to native mainland species. Ecological naiveté is defined as "the tendency for long-term inhabitants of low diversity and disharmonic (unbalanced) islands to lose their capacities for detecting, avoiding or otherwise coping with competitors, predators and parasites from the mainland or otherwise more-balanced and species-rich assemblages—i.e., species at least initially absent from these islands" (cited from (*Lomolino, 2016*:8). An often-cited example is the Falkland wolf (*Dusicyon australis*), initially even considered a feral domestic dog, because of its remarkable tameness and, among others, white on its tail tip, muzzle and lower limbs (*Clutton-Brock, 1977*), but this was dismissed on the basis of divergence time between this species and its ancestor (*Slater et al., 2009*; *Austin et al., 2013*). *Darwin (1839)* was struck by so much naivety, and foresaw this species' fast extinction, analogous to that of the dodo; indeed, the last individual was killed less than 50 years later.

Irrespective of the genetic and developmental background of the 'domestication syndrome', it has been noted that changes in pigmentation are one of the first traits to appear during the domestication of brown rats selected for tameness (*Trut, 1999*; *Trut, Oskina & Kharlamova, 2009*). Pigmentation changes in the form of increased occurrence of white patches of fur were also observed in a free-living population of wild house mice, *Mus musculus domesticus* (*Schwarz & Schwarz, 1943*), that were experimentally kept at an uninhabited barn, fed *ad libitum* and regularly handled in the course of 14 years (*Geiger, Sánchez-Villagra & Lindholm, 2018*). The repeated and frequent exposure to humans in combination with abundant food and absence of predators resulted in an indirect selection for tameness (*Geiger, Sánchez-Villagra & Lindholm, 2018*), with as it seems similar covariation of features of the 'domestication syndrome', in this case white fur patches and shorter snouts. Depigmentation might thus be a useful indicator for the detection of the early stages of evolutionary changes in murids which are typical for the 'domestication syndrome'. The 'domestication syndrome' might have a partial developmental similarity with the 'island syndrome' as far as increased naiveté (loss of anti-predator strategies) is concerned. If that is the case, one would expect increased depigmentation (especially white patches and brown regions) in natural insular populations on islands with the highest levels of insularity (low biodiversity, small size, high isolation). The opposite, increased pigmentation or melanism, is linked to insularity only in lizards, but the higher prevalence of melanistic colour morphs in more densely settled urban contexts in squirrels

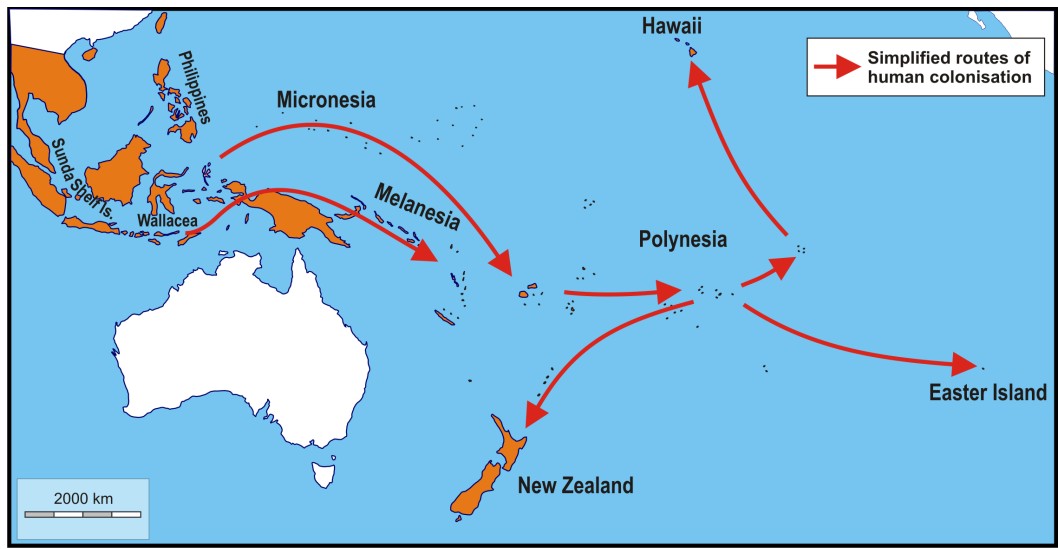

**Figure 1** **Schematic map showing simplified routes of human-aided dispersal of Polynesian rats, *Rattus exulans*.** Populations on the Sunda Shelf Islands were isolated since the Last Glacial Maximum when sea level rises disconnected these areas from each other and the mainland. Populations in the Philippines and Wallacea were introduced about 4,000–3,500 BP. Subsequently, more remote areas were reached starting about 3,400–3,200 BP (Micronesia, Melanesia). The remotest areas of Polynesia (New Zealand, Hawaii and Easter Island), finally, were settled about 820–720 years ago (*Roberts, 1991*; *Matisoo-Smith et al., 1998*; *Summerhayes et al., 2010*; *Wilmshurst et al., 2011*; *West et al., 2017*). Orange: native and introduced range. White: areas without Polynesian rats. Image credit: George Lyras. Map credit: https://d-maps.com/carte.php?num_car=3258&lang=en.

(*Garroway et al., 2018*) and ship rats (*Ondrias, 1966*), as well as in the latter species on the Galápagos Islands (*Patton, Yang & Myers, 1975*) and Hawaii (*Tomich, 1968*), suggests that both decreased and increased pigmentation may be expected in insular and/or commensal settings. Here, I examine this hypothesis on wild populations of the Polynesian rat, *Rattus exulans* (*Peale, 1848*) from the islands of the Pacific and insular Southeast Asia.

The Polynesian rat has vastly extended its geographic range through human-mediated transport (Fig. 1) starting c. 4,000 years ago (*Roberts, 1991*) and has been established in mainland Southeast Asia, the Greater and Lesser Sunda islands, the Moluccas, the Philippines, New Guinea, New Zealand and practically all inhabited Pacific islands for at least six centuries (*Corbet & Hill, 1992*; *Musser & Carleton, 2005*); additionally, it has been reported from Taiwan and Miyakojima of the southern Ryukyu Islands (arrival likely less than a century ago;(*Motokawa et al., 2001*) and Adele Island, northern Australia (arrival c. 130 years ago; (*Boyle et al., 2004*). On the islands outside its native range, it lives as a human commensal in houses, granaries, gardens, cultivated lands, plantations, anthropogenic scrubland, secondary forest and slightly altered primary forests near habitats maintained by humans (*Musser, 1977*; *Corbet & Hill, 1992*), where it is regularly exposed to humans. In this respect it is relevant to note that a number of historical accounts describe this rat as not shy towards humans (e.g., *Von Kotzebue, 1821* for the Marshall Islands) and easy to catch (*Peale, 1848*) for the Society Islands).
**Table 1 List of mainland countries and islands with number of Polynesian rats sampled in this study.**

| Region | Country or Island | Gender | | | | |
| --- | --- | --- | --- | --- | --- | --- |
| | | Female | Male | Unknown | Subtotal | Total |
| **Mainland** | | | | | | **370** |
| | Bangladesh | 2 | 0 | 0 | 2 | |
| | Burma | 3 | 5 | 0 | 8 | |
| | Laos | 3 | 2 | 0 | 5 | |
| | Malaysia | 6 | 5 | 0 | 11 | |
| | Myanmar | 19 | 16 | 0 | 35 | |
| | Thailand | 83 | 60 | 4 | 147 | |
| | Vietnam | 84 | 77 | 1 | 162 | |
| **Melanesia** | | | | | | **757** |
| | Bougainville | 38 | 47 | 0 | 85 | |
| | Efaté | 4 | 2 | 0 | 6 | |
| | Emirau | 1 | 4 | 0 | 5 | |
| | Espiritu Santo | 17 | 18 | 0 | 35 | |
| | Fergusson | 3 | 4 | 0 | 7 | |
| | Fiji (island N/A) | 2 | 1 | 1 | 4 | |
| | Goodenough | 11 | 12 | 0 | 23 | |
| | Grande Terre | 33 | 20 | 0 | 53 | |
| | Guadalcanal | 17 | 27 | 2 | 46 | |
| | Kiriwinia Island | 2 | 0 | 0 | 2 | |
| | Misima Island | 3 | 2 | 0 | 5 | |
| | New Britain | 15 | 15 | 1 | 31 | |
| | New Guinea | 152 | 198 | 30 | 380 | |
| | New Ireland | 2 | 2 | 0 | 4 | |
| | Normanby Island | 11 | 14 | 0 | 25 | |
| | Nusi Island | 0 | 1 | 0 | 1 | |
| | Ontong Java | 3 | 2 | 0 | 5 | |
| | Rossel Island | 9 | 10 | 0 | 19 | |
| | Ugi | 0 | 1 | 0 | 1 | |
| | Vanatinai | 4 | 9 | 0 | 13 | |
| | Woodlark Island | 4 | 3 | 0 | 7 | |
| **Micronesia** | | | | | | **292** |
| | And atoll | 1 | 0 | 0 | 1 | |
| | Arno atoll | 4 | 1 | 0 | 5 | |
| | Babelthuap Island | 3 | 1 | 0 | 4 | |
| | Bikar atoll | 6 | 6 | 1 | 13 | |
| | Bikini Atoll | 3 | 11 | 0 | 14 | |
| | Carlos Island | 1 | 2 | 1 | 4 | |
| | Caroline Islands (island N/A) | 1 | 1 | 0 | 2 | |
| | David Island | 1 | 0 | 0 | 1 | |
| | Dick Island | 1 | 0 | 0 | 1 | |
| | Enewetok Atoll | 1 | 1 | 0 | 2 | |

**Table 1** (*continued*)

| Region | Country or Island | Gender | | | Subtotal | Total |
|---|---|---|---|---|---|---|
| | | Female | Male | Unknown | | |
| | Guam | 20 | 41 | 0 | 61 | |
| | Guguan Island | 0 | 2 | 0 | 2 | |
| | Kalap Island | 1 | 0 | 0 | 1 | |
| | Kapingamarangi Atoll | 2 | 1 | 0 | 3 | |
| | Kayangel Atoll | 0 | 1 | 0 | 1 | |
| | Koror Island | 11 | 10 | 0 | 21 | |
| | Kosrae | 1 | 3 | 0 | 4 | |
| | Kwajalein Atoll | 1 | 0 | 0 | 1 | |
| | Majuro Atoll | 1 | 9 | 0 | 10 | |
| | Moen Island | 3 | 2 | 0 | 5 | |
| | Peleliu Island | 0 | 1 | 0 | 1 | |
| | Pohnpei | 23 | 9 | 0 | 32 | |
| | Ponam | 1 | 0 | 0 | 1 | |
| | Rongelap Atoll | 2 | 4 | 0 | 6 | |
| | Saipan | 7 | 1 | 0 | 8 | |
| | Taka Atoll | 0 | 2 | 0 | 2 | |
| | Tinian Island | 0 | 2 | 0 | 2 | |
| | Ulithi Atoll | 18 | 43 | 0 | 61 | |
| | Wake Island | 8 | 14 | 0 | 22 | |
| **Polynesia** | | | | | | 424 |
| | Atafu Atoll | 2 | 0 | 0 | 2 | |
| | Aunu'U Island | 2 | 5 | 0 | 7 | |
| | Birnie Atoll | 6 | 18 | 0 | 24 | |
| | Fakaofo Atoll | 1 | 1 | 0 | 2 | |
| | Hatutaa Island | 2 | 2 | 0 | 4 | |
| | Hawaii | 18 | 16 | 1 | 35 | |
| | Hen Island | 1 | 3 | 0 | 4 | |
| | Kiritimati | 0 | 1 | 5 | 6 | |
| | Kanton | 6 | 2 | 2 | 10 | |
| | Kauai | 1 | 0 | 0 | 1 | |
| | Kure Atoll | 25 | 36 | 1 | 62 | |
| | Little Barrier | 18 | 17 | 0 | 35 | |
| | Manra Island | 1 | 0 | 0 | 1 | |
| | Macauley | 1 | 1 | 0 | 2 | |
| | Maui | 7 | 6 | 0 | 13 | |
| | North Island* | 6 | 7 | 73 | 86 | |
| | Niuafo'Ou Island | 1 | 2 | 0 | 3 | |
| | Nukunonu Atoll | 1 | 1 | 0 | 2 | |
| | Oahu | 13 | 13 | 1 | 27 | |
| | Ofu | 7 | 3 | 0 | 10 | |
| | Olosega | 2 | 2 | 0 | 2 | |
| | Onotoa Atoll | 2 | 3 | 1 | 6 | |
| | Orona Atoll | 1 | 1 | 1 | 3 | |

**Table 1** (*continued*)

| Region | Country or Island | Gender | | | | Total |
|---|---|---|---|---|---|---|
| | | Female | Male | Unknown | Subtotal | |
| | Pitcairn | 0 | 3 | 1 | 4 | |
| | Raoul | 6 | 1 | 1 | 8 | |
| | Raroia Atoll | 4 | 3 | 0 | 7 | |
| | Repanga | 2 | 5 | 0 | 7 | |
| | Rose Atoll | 3 | 5 | 0 | 8 | |
| | Rurima Rocks | 0 | 0 | 4 | 4 | |
| | South Island | 1 | 2 | 3 | 6 | |
| | Stewart | 0 | 2 | 0 | 2 | |
| | Swains Island | 7 | 4 | 0 | 11 | |
| | Tahiti | 0 | 0 | 1 | 1 | |
| | Ta'u Island | 0 | 1 | 0 | 1 | |
| | Tepoto | 1 | 0 | 0 | 1 | |
| | Vostok Island | 5 | 5 | 0 | 10 | |
| | Whakaari | 2 | 5 | 0 | 7 | |
| **Sunda Shelf Islands** | | | | | | **478** |
| | Bali | 2 | 5 | 0 | 7 | |
| | Borneo | 75 | 55 | 3 | 133 | |
| | Borneo, Pulo Balik Kukup | 10 | 16 | 0 | 26 | |
| | Java | 53 | 88 | 0 | 141 | |
| | Palawan | 25 | 27 | 5 | 57 | |
| | Palawan, Balabac Island | 11 | 11 | 0 | 22 | |
| | Culion | 3 | 2 | 0 | 5 | |
| | Sumatra | 19 | 25 | 0 | 44 | |
| | Sumatra, Banka Island | 1 | 0 | 0 | 1 | |
| | Sumatra, Batam Island | 4 | 0 | 0 | 4 | |
| | Sumatra, Nias Island | 4 | 1 | 0 | 5 | |
| | Sumatra, Simeulue | 5 | 17 | 0 | 22 | |
| **Wallacea** | | | | | | **722** |
| | Ambon | 1 | 1 | 0 | 2 | |
| | Basilan Island | 0 | 3 | 0 | 3 | |
| | Buru | 0 | 1 | 0 | 1 | |
| | Busuanga Island | 8 | 9 | 0 | 17 | |
| | Catanduanes | 1 | 0 | 0 | 1 | |
| | Halmahera | 10 | 20 | 2 | 32 | |
| | Leyte | 1 | 0 | 1 | 2 | |
| | Luzon | 104 | 104 | 10 | 218 | |
| | Malenge Island | 11 | 14 | 0 | 25 | |
| | Mindanao | 24 | 15 | 4 | 33 | |
| | Mindoro | 20 | 23 | 2 | 45 | |
| | Negros | 11 | 18 | 2 | 31 | |
| | Panay | 4 | 1 | 0 | 5 | |

**Table 1** (*continued*)

| Region | Country or Island | Gender | | | Subtotal | Total |
|---|---|---|---|---|---|---|
| | | Female | Male | Unknown | | |
| | Peleng Island | 9 | 4 | 0 | 13 | |
| | Polillo Island | 0 | 0 | 1 | 1 | |
| | Samar | 4 | 5 | 0 | 9 | |
| | Sulawesi | 118 | 147 | 0 | 265 | |
| | Taliabu | 9 | 9 | 1 | 19 | |
| **Total** | | | | | | **3,043** |

Evolutionary divergence is expected to be greatest in the following populations: those on islands with reduced competitor and predator stress through relaxed selection (*Whittaker & Fernandez-Palacios, 2007*; *Lomolino, Riddle & Whittaker, 2017*), those on small islands (*Lomolino et al., 2012*; *Van der Geer et al., 2018*) and those with greater residence times (*Lomolino et al., 2013*; *Van der Geer, Lomolino & Lyras, 2018*). The hypothesis is that these populations of Polynesian rats will show the highest incidence of aberrant coat patterns.

## MATERIALS & METHODS

Dried and prepared skins of the Polynesian (or Pacific) rat, *Rattus exulans*, were studied at the American Museum of Natural History (New York, US), the Smithsonian Institution (Washington, US) and Te Papa Tongarewa National Museum of New Zealand (Wellington, NZ) (for specimen numbers, see Data S1). Total number of specimens was 3,034, of which 370 originated from a total of 7 mainland countries (see Table 1 and Data S1 for details). The islands are classified in five biogeographical regions that differ in biodiversity and average colonisation period by the focal species. Numbers of island specimens are as follows per group: Melanesia ($n = 757$ from 21 islands), Micronesia ($n = 292$ from 29 islands), Polynesia ($n = 424$ from 37 islands), Sunda Shelf Islands ($n = 478$ from 12 islands) and Wallacea ($n = 722$ from 18 islands). Palawan and its satellites are considered here as part of the Sunda Shelf, following *Cranbrook Earl of (2000)*, who maintains a connection with Borneo during the Last Glacial Maximum (see, however, *Voris (2000)* who dismisses any connection with Borneo during the Late Pleistocene, based on sea-level reconstructions). The remainder of the Philippines are oceanic in origin, except for Mindoro, which originated as a portion of the Asian continental shelf, and have never been connected to a mainland (*Van der Geer et al., 2010*). They are here considered together with Wallacea on grounds of its similar isolation from the Asian and Australian continental shelves. Bali was connected to Java during the Late Pleistocene (*Voris, 2000*), and is thus classified with the Sunda Shelf Islands. Offshore islands are considered together with the respective main island.

The biogeographical regions differ in the composition of their faunas (see Data S2) for number of ecologically relevant native competitors and predators, based on fauna lists provided by *Van der Geer, Lomolino & Lyras (2017)* and *Van der Geer, Lomolino & Lyras (2018)*. For example, while Sunda Shelf islands (both Greater and Lesser Sunda Islands) and parts of Melanesia have native competitors and predators, Wallacea has only

native competitors, whereas Polynesia and Micronesia have no competitors or predators at all. Introduced potentially relevant predators consists of feral cats in this case. Their impact on population dynamics of introduced rats is, however, negligible (*Parsons et al., 2018*) and their presence is not taken into account here. Time in isolation is taken from (*Kirch, 1984*; *Roberts, 1991*; *Matisoo-Smith et al., 1998*; *Cranbrook Earl of, 2000*; *Anderson & White, 2001*; *Anderson & Sinoto, 2002*; *Anderson, 2005*; *Anderson, 2008*; *Green, Jones & Sheppard, 2008*; *Summerhayes et al., 2010*; *Hanebuth et al., 2011*; *Wilmshurst et al., 2011*; *Burley, Weisler & Zhao, 2012*; *Denham, Ramsey & Specht, 2012*; *Carson et al., 2013*; *Clark et al., 2016*; *McCoy et al., 2016*; *West et al., 2017*) (details per island provided in Data S2; see also Fig. 1) and binned in classes of 1,000 years each, except for the highest class (class 5), which represents time in isolation exceeding 5,000 years without upper limit. Surface area was $log_{10}$ transformed before statistical analyses.

Aberrant coat colour and patterns are those that differ substantially (evaluated by me and, where needed, checked by the photographer and curatorial staff) from those seen in the rest of the population from the same island or country, thus taking natural variation within populations into account. Both upper- and undersides are taken into account. Slightly darker or lighter coat colours are considered normal, taking seasonal variations into account. Dorsal pelage in *R. exulans* varies per region between grey-brown to reddish-brown; ventral pelage is dark grey or pale buffy with pale grey or white tips (*Tate, 1935*; *Atkinson & Moller, 1990*). White-bellied individuals, lacking grey bases to the white belly hair, are typical for and restricted to Flores (Lesser Sunda Islands; Indonesia) (*Schwarz & Schwarz, 1967*) and Adele Island (Australia) (*Tate, 1951*); the mtDNA haplotype (Rx083) is the same in the Flores sample from Liang Bua cave and Adele Island, while different elsewhere (*Hingston, 2015*), supporting a recent introduction by Indonesian fishermen.

I here take the mainland populations as reference, and not those from the Lesser Sunda Islands. Flores was indicated as the place of origin of the *exulans* group (*Schwarz, 1960*), instead of mainland Southeast Asia (*Tate, 1935*), based on the presence here of white-bellied forms. *Schwarz & Schwarz (1967)* considered these the "wild form" and the ancestor of all commensal types. They interpreted the loss of a white belly as an adaptation to the darker indoor environment. Genomics confirmed this hypothesis was (*Thomson et al., 2014*). Note, however, that *Musser (1981)* did not find significant morphological differences between *R. exulans* from Flores and the rest of the Lesser Sunda Islands; in addition, no fossil or subfossil *R. exulans* have yet been reported from Flores, despite the rich murid fossil record with stratigraphic contexts. Although the exact place of origin has important implications for tracing human migrations through insular Southeast Asia, it has much less relevance to the distribution of coat polymorphisms.

Aberrant coat patterns are here defined in terms of leucism and melanism. Leucistic mammals lack pigment to a varying degree, ranging from a few white spots to (nearly) all white or yellowish. In contrast to albinistic mammals, that congenitally lack pigment altogether, eyesight is not usually impaired and the nose, eyes and feet are coloured as typical for the species. Melanistic mammals in contrast show an increased expression of black or brown pigment (melanine). Mixed white and brown/black animals are generally known as piebald.

Potentially, three types of sampling biases could have influenced the results presented here. Particular collectors may have either preferred or discarded specimens with an aberrant or otherwise remarkable coat pattern. Particular curators may have discarded particular aberrant specimens during curatorial activities. Finally, sample sizes are not equal for all regions: the samples from Wallacea and Melanesia are twice the size of those from the other three island regions as well as the mainland, potentially increasing the chance of finding abnormalities in the first two regions. I here assume the collector's and curatorial bias to be similar across the regions. Furthermore, the specimens are derived from only three major collections, limiting the curatorial bias to differences among these collections. Overlap in geographical provenance between the collections further reduces this bias.

The variables (number of competitors and predators, island area, time in isolation and cases of aberrant coat patterns) were not normally distributed (after testing using the Shapiro–Wilk test; *Shapiro & Wilk, 1965*; Table S1), therefore Chi-square tests were used to test whether the aberrant coat patterns were evenly distributed over the archipelagoes and the time bins, and whether there was a relation with the number of native predators and competitors. The groupings were based on biogeographical provinces (Mainland, Melanesia, Micronesia, Polynesia, Sunda Shelf Islands, Wallacea), as this corresponds best to biodiversity and residence time (see above), absence/presence of aberrant coat patterns and residence time (binned in steps of 1,000 years each, see above). The number of native predators was binned as zero, 1, 2 or more predators; the number of native competitors was binned as zero, 1–5, 6–30 or more competitors (see Data S2 for the islands; mainlands fall in the latter bins). Regarding island area ($\log_{10}$ transformed), a non-parametric Kruskal–Wallis one-way analysis of variance test (by lack of normality) was used to determine if there was a correlation between coat colour polymorphism (*Siegel, 1956*), with the number of aberrant coat patterns as grouping variable (three samples: 0, 1 or 2 aberrant coats) and island area, and similarly, to test for a correlation between island area and number of competitors. Statistical analyses were performed using Past 3.16 (*Hammer, Harper & Ryan, 2001*). Significance level alpha = 0.05.

## RESULTS

Out of 3,034 skins of *R. exulans*, 15 skins showed aberrant coat colour and/or pattern, evenly distributed over the sexes (Tables 2 and 3) and over the geographic distribution (Fig. 2); for summary statistics, see Table S1. The single aberrant coat pattern from the mainland (Vietnam) is an albino (Fig. 3A, 3B). Albinos are, however, characterised by a congenital failure to produce melanin in otherwise normal melanocytes. In this way, potential changes in the coat colour pattern are not expressed and can thus not be traced. Albinos can further be recognised by their red eyes and pink noses and ears, as a result of this total lack of pigmentation. This specimen is omitted from the analyses. Note that albino specimens are reported in less than 2% of rodent species (*Romero, Racines-Márquez & Brito, 2018*), and that the specimen shown here (USNM 357550) is, by my knowledge, the first reported case in *R. exulans*.

van der Geer (2019), PeerJ, DOI 10.7717/peerj.6894

**Table 2  Polynesian rats with an aberrant coat pattern.** Specimens are first ordered according to the biogeographic region and then on specimen number. The number of ecologically relevant native predators and competitors and island area (km$^2$) are from Van der Geer, Lomolino & Lyras (*2017*, *2018*), except for Batam (*Meijaard, 2003*) and Malenge (*Laurie & Hill, 1954*). Area is rounded to the nearest integer.

| Region | Country | Specimen | | | | sex | aberrant coat pattern |
|---|---|---|---|---|---|---|---|
| Mainland | Vietnam | USNM | 357550 | | | m | albino |
| **Region** | **Island** | **Area** | | **Predators** | **Competitors** | | |
| Micronesia | Saipan | 123 | USNM 277708 | 0 | 0 | f | hazelnut colour (dark buff) |
| | Pohnpei | 334 | USNM 301999 | 0 | 0 | f | melanistic, also underside |
| Polynesia | Hawaii | 10,435 | USNM 324887 | 0 | 0 | f | piebald, mixture of white and brown, also at underside |
| | Kure | 1 | USNM 338630 | 0 | 0 | f | very dark, toward melanistic |
| | Kauai | 1,435 | USNM 570868 | 0 | 0 | f | very dark, toward melanistic; few white hairs |
| | Kiritimati | 321 | USNM 338688 | 0 | 0 | m | ash grey, toward leucism |
| Sunda Shelf Islands | Batam | 399 | USNM 143251 | 0 | 0 | f | blackish patches, mainly on head and shoulder and median ridge of back; very clear at underside of mandible, left cheek and right front leg |
| | Java | 138,794 | USNM 481593 | 13 | 7 | m | white spot between the shoulders |
| Wallacea | Malenge | 13 | AMNH 153296 | 0 | 1 | m | buff coloured, large irregular patches at underside |
| | Sulawesi | 180,681 | AMNH 153015 | 1 | 36 | f | white patches on both shoulders |
| | Sulawesi | 180,681 | AMNH 153037 | 1 | 36 | f | white patch sagitally on back |
| | Luzon | 109,965 | USNM 145823 | 0 | 40 | m | white patch below left shoulder |
| | Luzon | 109,965 | USNM 145827 | 0 | 40 | f | white patch at left shoulder |
| | Mindoro | 10,572 | USNM 278589 | 0 | 6 | m | leucism: buff coloured, slightly darker than albino, underside almost white ("aberrant color" on label) |

**Table 3   Percentage of aberrant coat patterns of Polynesian rats per region.** The time spent is an average value, estimated based on the majority of the relevant islands (see Fig. 1). Abnormalities are the combined cases of melanism and leucism. The albino specimen from the Mainland is here excluded (see Results section).

| Region | Time spent in situ (ka) | Number of specimens | Abnormalities | | | |
|---|---|---|---|---|---|---|
| | | | total | % | female | male |
| Mainland | 2,200 | 370 | 0 | 0 | 0 | 0 |
| Melanesia | 3.4–3.2 | 757 | 0 | 0 | 0 | 0 |
| Micronesia | 1.75 | 292 | 2 | 0.68 | 2 | 0 |
| Polynesia | 0.80 | 424 | 4 | 0.94 | 3 | 1 |
| Sunda Shelf | 12 | 478 | 2 | 0.42 | 1 | 1 |
| Wallacea | 4–3.5 | 722 | 6 | 0.83 | 3 | 3 |
| **Total** | | **3,043** | **14** | **0.46** | **7** | **5** |

The aberrant coat colour patterns from the islands by contrast either show signs of increased pigmentation (melanism) or reduced pigmentation (leucism). The depigmentation can be complete (Figs. 3C 3D; Fig. S1 A), partial (piebald; Fig. 4) or restricted to a single spot (Fig. 5, Fig. S1B) or a few strands of hair (Fig. 2I). The piebald specimen (Hawaii) has normal, wild-type colouration on the head, shoulders and a wide mid-dorsal stripe or band extending to the base of the tail, as typical for the "hooded" phenotype of laboratory rats (*R. norvegicus*); different from the hooded rats is the patchy brown pigmentation on the white parts. Based on the typical light buff colour (pale yellow) of the albino *R. exulans* (Figs. 3A–3B) as a result of total lack of melanin in the melanocytes, I here interpret the evenly buff coloured specimens as different phases of leucism with differential expression of melanin, where a total lack corresponds to the colour of the albino (but with different congenital background) (Figs. 3C–3D; Fig. S2A). The latter specimens are not albinos because they have black eyes, coloured ears and toe pads, and presumably black eyes (eyes were removed, but aberrant features are typically noted down, which was not the case here). The melanistic specimens are either evenly darker coloured, including (Figs. 6A–6B) or excluding the underside (Figs. 6C–6D; Fig. S2B–Fig. S2C), or bear irregular patches of increased pigmentation, either on both upper- and underside (Figs. 6E–6F) or only at the underside (Figs. 6G–6H).

The difference in distribution of aberrant coat colour patterns over the six biogeographical regions is not significant (Chi$^2$, $p = 0.08$, all patterns; $p = 0.06$, melanism and leucism considered separately; Table S2), nor is the distribution different between the mainland and all islands considered together (Chi$^2$, $p = 0.162$). Within Polynesia, coat polymorphisms are restricted to the Hawaiian archipelago and Line Islands, both at the north-eastern limit of the distribution. The majority of aberrant coat patterns are various phases of leucism.

Time in isolation cannot predict the occurrence of aberrant coat patterns (Chi$^2$, $p = 0.673$), whereas the numbers of native predators and native competitors as well as log transformed island area are significantly correlated to the number of aberrant coat patterns (Chi$^2$, $p = 0.017$, Chi$^2$, $p = 0.002$ and Kruskal–Wallis, $p < 0.001$, respectively).

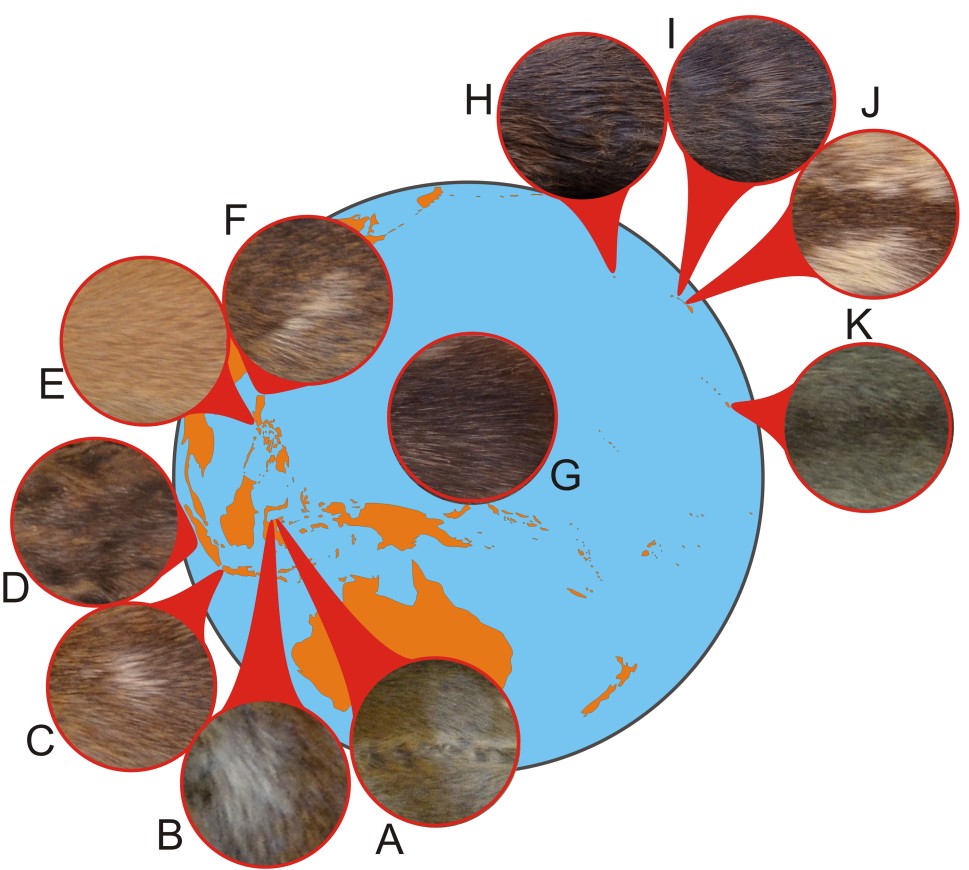

**Figure 2** **Schematic map showing the distribution of aberrant coat patterns in Polynesian rats.** The distribution of coat polymorphisms is random over the introduced range. No aberrant coat patterns were found in populations from the native range (mainland Southeast Asia), except for an albino specimen from Vietnam (not indicated here; see Fig. 3). The circles show details of ventral (A) and dorsal skins (B); all skins are shown in full in Figs. 3–6 and Figs. S1–S2. (A) Malenge, Togian Islands, Sulawesi. (B) Sulawesi, Indonesia. (C) Java, Indonesia. (D) Batam, Riau Islands, Sumatra. (E) Mindoro, Philippines. (F) Luzon, Philippines. (G) Pohnpei, Caroline Islands. (H) Kure atoll, Hawaii. (I) Kauai, Hawaii. (J) Hawaii, Hawaii. (K) Kiritimati. For specimen details, see Table 2. Image credit: George Lyras.

The latter two variables are, however, positively correlated to the number of aberrant coat patterns. This counterintuitive result is probably driven by the positive correlation between island area and competitors (species–area relationship; Kruskal–Wallis, $p < 0.001$), and the correlation coat polymorphism and island area is here considered redundant. Test results are provided in Tables S1 and S2.

## DISCUSSION

Insular populations of Polynesian rats show a higher incidence of individuals with an aberrant coat pattern than those of mainland Polynesian rats, which lacked leucistic and/or melanistic individuals altogether. This higher incidence is, however, not statistically significant, perhaps as a result of the smaller sample size of mainland rats (12% of the total). The coat polymorphisms found on the islands are not significantly correlated with time in

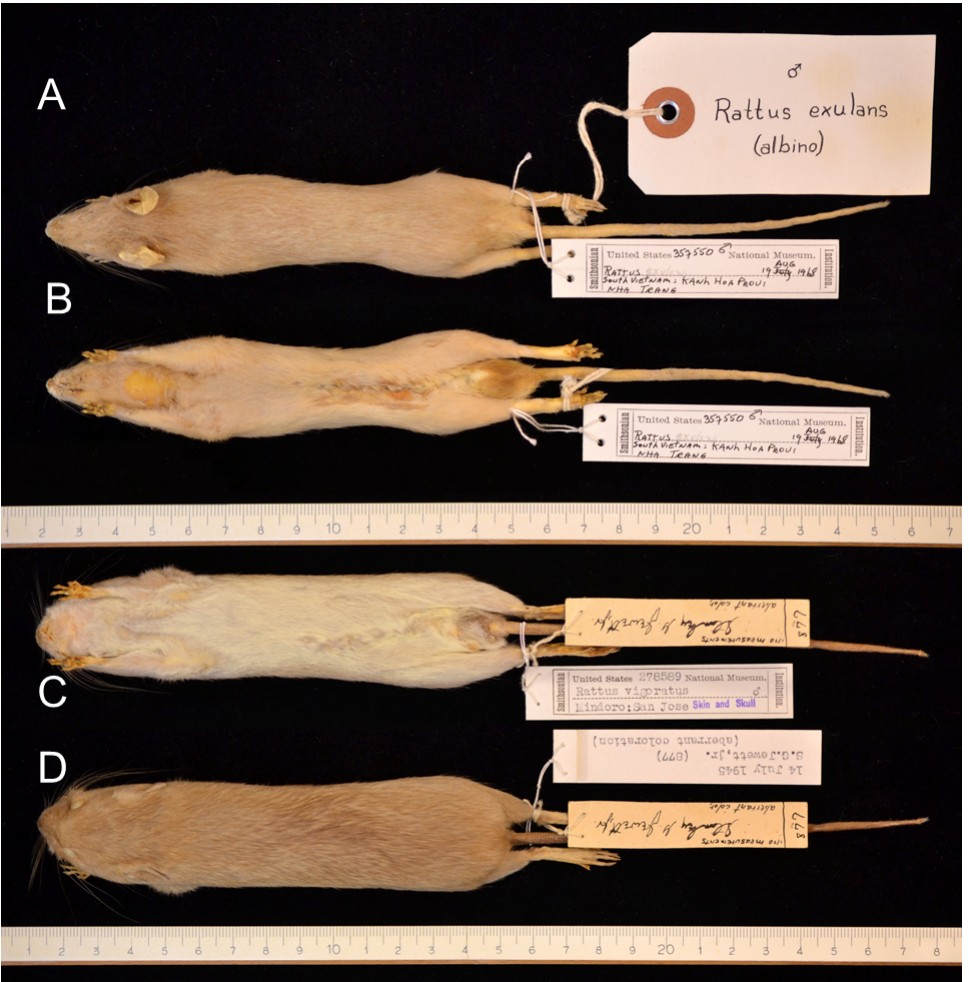

**Figure 3** **Albino (A, B) and complete leucistic (C, D) coat patterns in the Polynesian rat.** (A) Albino specimen from Vietnam (USNM 357550, male), dorsal view. (B) Same specimen, ventral view. (C) Light buff coloured (complete leucism) specimen from Mindoro, Philippines (USNM 278589, male), ventral view. Note that *Rattus vigoratus* as written on the label is a junior synonym for *Rattus exulans* (Polynesian rat), following *Wilson & Reeder (2005)*. (D) Same specimen, dorsal view. Photo credits: Bartholomeus van der Geer.

isolation. Indeed, although the Philippines and Sulawesi were among the first islands to be reached by the Lapiti people on their way into the Pacific some 3,500–4,000 years ago (*Anderson, 2005*), and half the reported cases are from these islands, an equal amount of cases of aberrant coats is from Polynesian islands that were reached only some 925–657 years ago (*West et al., 2017*). Colour polymorphisms seem not to have been facilitated by the sequential colonisation (the founder effect) either, as seen by the lack of a clear distribution pattern of (nearly) identical aberrant coat patterns (Figs. 1 and 2). However, note that within Polynesia, the aberrant cases are restricted to the most remote part of the north-eastern distribution (Hawaii and Line Islands). A sequence of founding events, such as is the case for the Polynesian dispersal from Near to Remote Oceania is likely to

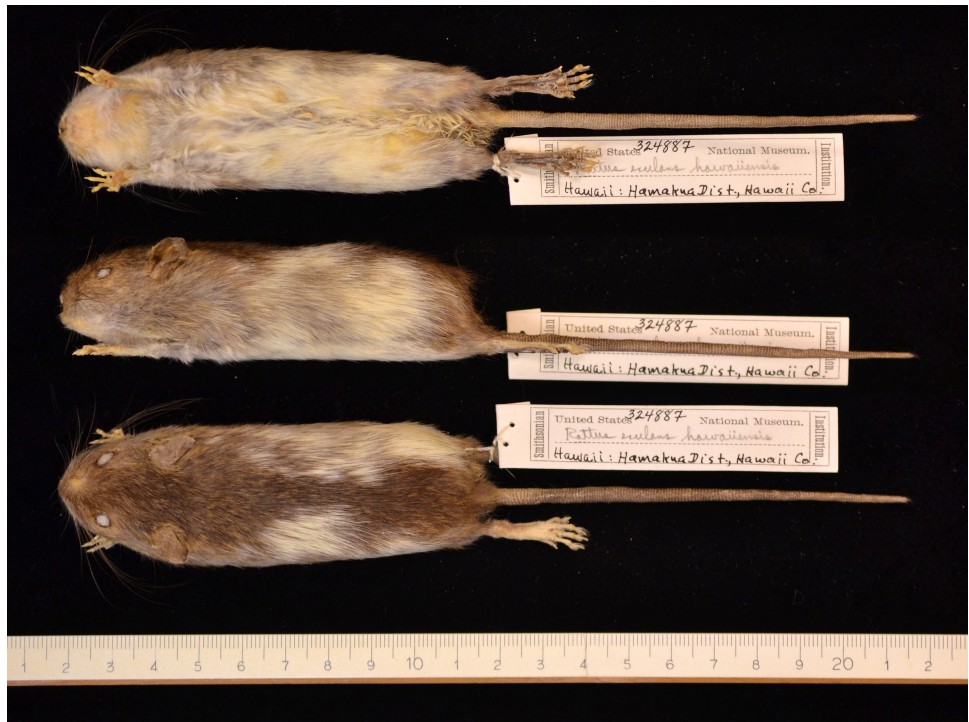

**Figure 4  Piebald coat pattern in an insular Polynesian rat.** The piebald pattern (a mixture of brown and white) is present on the ventral, lateral and dorsal sides in this specimen from Hawaii (USNM 324887, female). Photo credits: Bartholomeus van der Geer.

reduce genetic variation (e.g., *Clegg et al., 2002*). However, this cannot explain the results presented here, as similar polymorphisms would be expected to occur on related islands, which is not the case.

The level of predation and murid competition varies substantially between the regions. Whereas Polynesia and Micronesia both lack ecologically relevant native predators and competitors, Wallacea and the Sunda Shelf islands house a suite of native murids to compete with, as well as native predators in the case of the latter islands. The unexpected positive correlation between coat colour polymorphisms and amount of ecologically relevant competitors might be an artefact instead of representing a causal relationship, because these results are strongly influenced by the islands Luzon and Sulawesi, each with two aberrant specimens as well as an exceptionally high number of native murids. A simple mechanism that could explain the observation that the more competitors, the more aberrant coat patterns are present, is that potential expansion of Polynesian rats into non-commensal niches is hampered by resident murid species. I can here only speculate that this obligatory restriction to the human commensal niche, and the inherent reduced fear for humans, caused the higher number of abnormalities on (very) biodiverse islands. For Sulawesi and Luzon, there are indeed indications that this might be valid: its occurrence on Sulawesi was recorded mostly in and around villages, and only sparsely in secondary forest (*Musser, 1977*) and on Luzon it is considered always associated with human habitation,

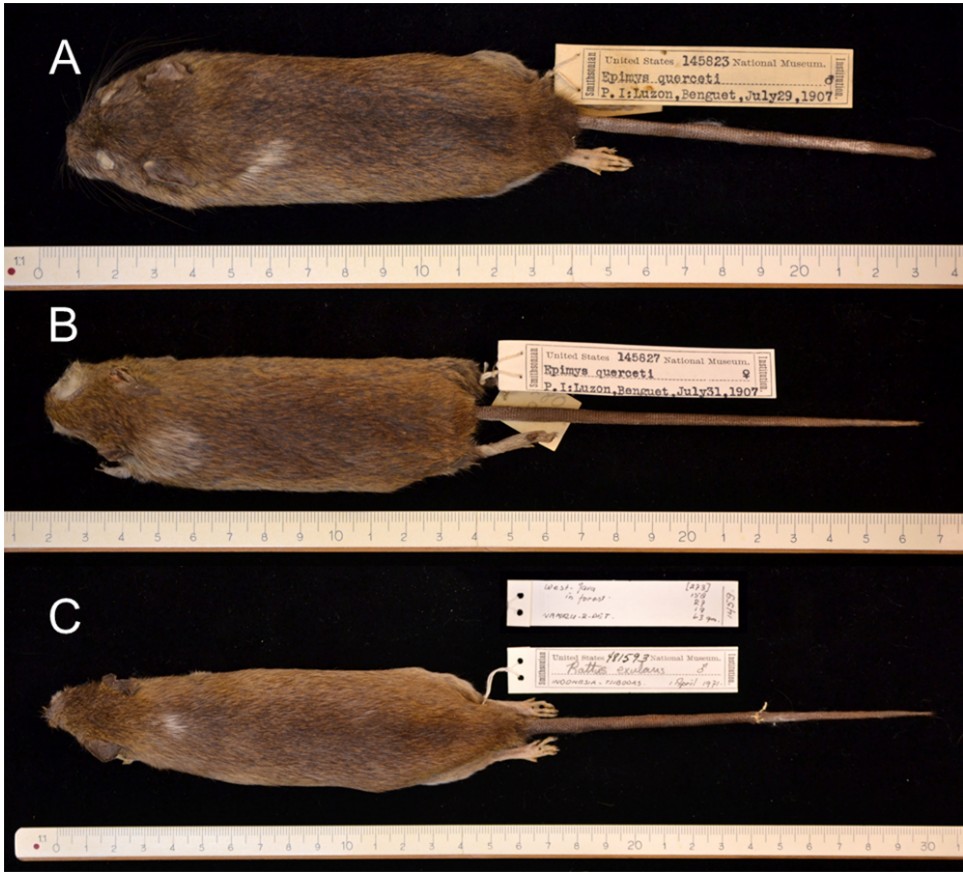

**Figure 5** **Insular Polynesian rats showing white patches at the shoulder region.** (A) Specimen from Luzon, Philippines (USNM 145823, male). Note that *Epimys querceti* as written on this label and the next is a junior synonym for *Rattus exulans* (Polynesian rat), following *Wilson & Reeder (2005)*. (B) Specimen from the same locality (USNM 145827, female). (C) Specimen from Java, Indonesia (USNM 138794, male). Photo credits: Bartholomeus van der Geer.

except when there are few native murids (*Heaney et al., 2010*). Similar scenarios have been suggested for introduced house mice, which are usually outcompeted by other rodent species in non-anthropogenic habitats (e.g., *Berry, Cuthbert & Peters, 1982*).

Depigmentation in the form of white patches on the trunk as presented by Polynesian rats from Java, Hawaii, Kauai, Luzon and Sulawesi has been observed in practically all domestic animal species (mouse, rat, guinea pig, rabbit, dog, cat, fox, mink, ferret, pig, reindeer, sheep, goat, cattle, horse, camel, alpaca and guanaco; (*Darwin, 1868*; *Belyaev & Trut, 1989*; *Gariépy, Bauer & Cairns, 2001*; *Trut, Oskina & Kharlamova, 2009*). In the brown rat, *R. norvegicus*, typically, the irregular white patches occur in particular sites: just below the throat and above the eyes, the paws and the tip of the tail. This is, however, not what we observe in the Polynesian rats, where the patches are predominantly located in the shoulder region and on the back of the thorax. Only the piebald specimen from Hawaii has an irregular placement of white patches all over its body and skull with the exception of the midsagittal region over its entire length, approaching observed colour patterns in domestic

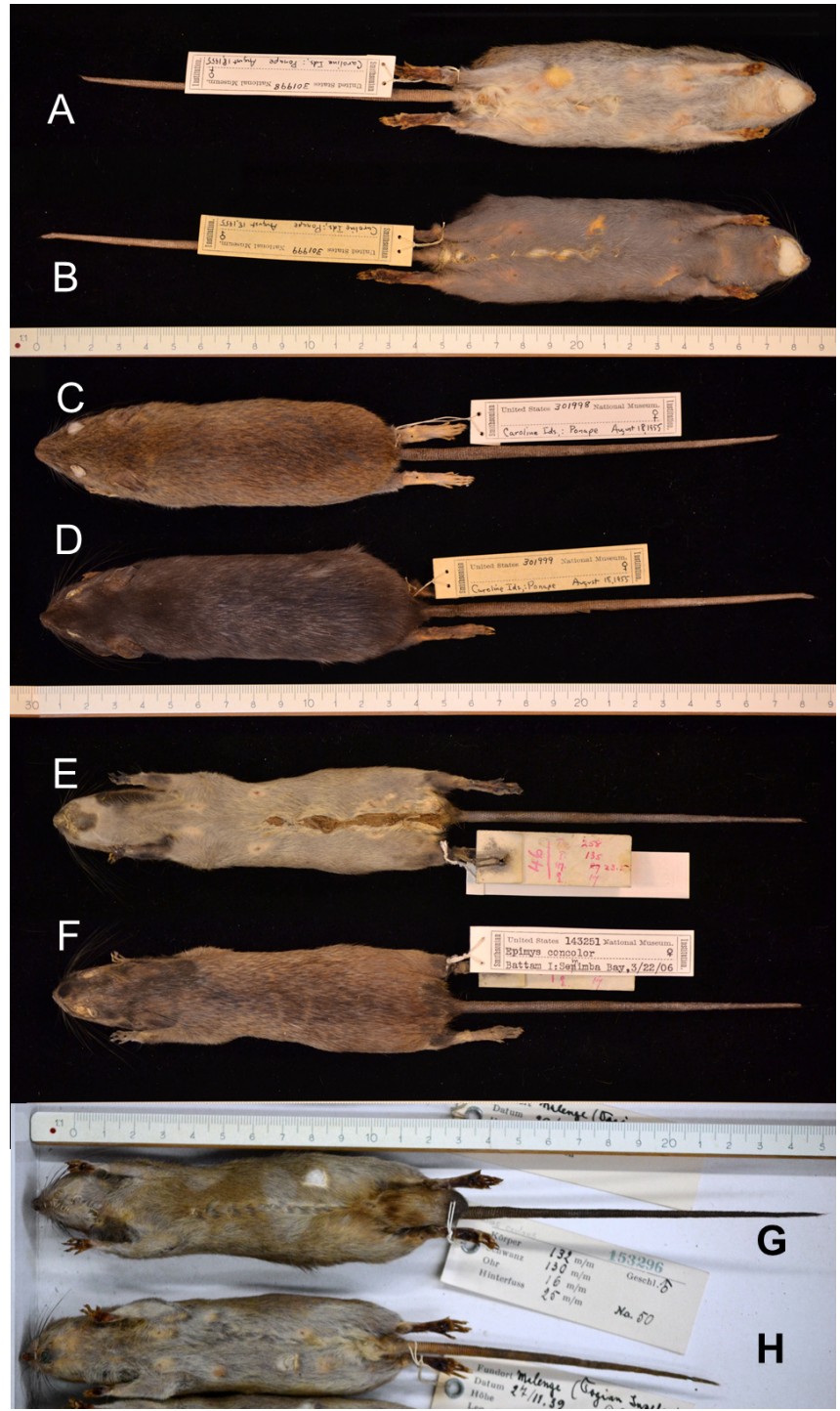

**Figure 6 Insular Polynesian rats with different expressions of melanism (increased pigmentation).**
(A) Normal-coloured specimen from Ponape, Caroline Islands (USNM 301998; female), ventral side. (B) Melanistic coloured specimen from the same population (USNM 301999, female), captured at the same day, ventral side. (C) Same specimen as (A), dorsal view. (D) Same specimen as (B), dorsal view. (E) Irregular black patches in a specimen from Batam Island, Sumatra (continued on next page...)

**Figure 6 (…continued)**
(USNM 143251, female), ventral view. (F) Same specimen, dorsal view. Patches are present on both upper- and underside. Note that *Epimys concolor* as written on the label is a junior synonym for *Rattus exulans* (Polynesian rat), following *Wilson & Reeder (2005)*. (G) Irregular dark buff patches in a specimen from Malenge, Togian Islands, Sulawesi (USNM 153296, male), ventral side. (H) Normal-coloured specimen (USNM 153295, female) from the same locality. Photo credits: Bartholomeus van der Geer.

brown rats. White patches in rats are generally considered as evolved along the commensal pathway of domestication (*Sánchez-Villagra, Geiger & Schneider, 2016*), and the Polynesian rat seems to follow this pattern, being a human commensal species in its introduced range, except for islands abandoned by the Polynesians centuries ago (e.g., the Mystery Islands). This pathway is characterized by unintentional selection for tameness (*Zeder, 2012a*; *Zeder, 2012b*; *Larson & Fuller, 2014*). In the case of island evolution, this tameness may also be a case of ecological naivité driven by the decreased number of predators and competitors relative to the mainland (e.g., the number of native mammalian predators feeding on rats of Vietnam is 24; data from (*IUCN, 2019*). Indeed, a significant correlation was found between native predators and incidence of aberrant coat patterns. This is best illustrated by the presence of the most extreme aberrant coat pattern (the piebald specimen) on an island (Hawaii) without any native mammals, except for bats.

In general, evolutionary changes of island mammals in their new environment may have been adaptive as well as non-adaptive (e.g., *Whittaker & Fernandez-Palacios, 2007*). In that light, a possible scenario for the higher occurrence of melanistic coats on Pohnpei, Kure Atoll and Kauai (Hawaii) could be that darker coat patterns have an adaptive advantage in avoiding birds of prey because such a colour spectrum corresponds to the background colour of their habitats: dark basaltic rocks covered in rain forests. However, the lack of increased melanism on other basaltic islands fails to confirm such a pattern. The lack of a clear pattern and the overall low percentage of melanism seems unable to support a functional explanation. In addition, although piebald coat patterns have the potential to be adaptive in rainforest environments (Hawaii), the high incidence of piebald patterns in domestic animals rather favours an explanation towards a higher degree of tameness under low predatory pressure.

The observed colour polymorphisms here are limited to variations in the expression of red-yellow and/or black-brown (distribution, expression, intensity). These two colour ranges in vertebrates are mainly controlled by two genes, being the melanocortin-1 receptor gene (*Mc1r*) and its antagonist, the *Agouti* signalling protein gene (*Asip*) (*Hubbard et al., 2010*). These genes work together in controlling the pigment expression of eumelanin (black-brown) and pheomelanin (red-yellow) in melanocytes. Melanistic furs in natural populations are most commonly associated with mutations in *Mc1r* (e.g., (*Mundy et al., 2004*), In mice, *Agouti* further regulates the differences between dorsal and ventral pigmentation and may be responsible for the switching off of pigment production altogether (*Vrieling et al., 1994*). Dark buff or rufous undersides (contrary to the usually white ventral pelage) were observed in the most remote population of the large Japanese field mouse, *Apodemus speciosus* (*Temminck, 1844*), on the Izu Islands (*Tomozawa et al., 2014*), explained as the result of loss of genetic variation and changes in the frequency of

an allele for a mutation in *Asip*. Laboratory mice with such mutations showed enhanced pigmentation (yellow pheomelanin) of the ventral pelage (*Vrieling et al., 1994*), similar to the pattern (dark buff with blackish tail) observed in the large Japanese field mouse on Miyake island (*Tomozawa et al., 2014*). Such a mechanism may explain the Polynesian rat specimen with irregular dark buff patches especially at its underside. White spotting in rats and mice, including hooding, is caused by mutations in one or more of the genes that regulate neural crest-derived melanoblast development (e.g., *Kit*, *Pax3*, *Mitf* and *Sox10*), leading to reduction and/or loss of mature melanocytes (*Baxter et al., 2004*; *Kuramoto et al., 2012*). White spots can be associated with disorders in other neural crest derived tissues or elsewhere due to pleiotropic effects, including microphthalmia, deafness, spina bifida and sterility (*Baxter et al., 2004*), most of which would be fatal or at least severely impact fitness in any wild living rat population. The pale-yellow coat pattern in the Polynesian rat might be explained by the diluting effects of the gene(s) that control the intensity of pigment, in combination with alleles coding for brown instead of black. In mice, this combination is the result of homozygosity for the recessive allele in both genes (*Griffiths et al., 2000*). In parallel, the ash grey dorsal fur of the Kiritimati specimen might be due to reduced pigment production in combinations with a black colour. In short, a number, or perhaps all, of the aberrant coats in Polynesian rats might be explained simply by loss of genetic variation, due to drift and founder effects.

Whereas phases of leucism, including piebald, can all be ascribed to homozygosity for recessive alleles in one or more genes (see above), this is not always the case for melanism. In laboratory mice, for example, mutations in alleles of *Mc1r* leads to dominant melanism, whereas mutations in alleles of *Agouti* usually cause recessive melanism. The latter was found in four geographically widely separated North American deer mouse populations, which suggests that melanism-inducing mutations in *Agouti* might be more common than generally thought (*Kingsley et al., 2009*). The low incidence (three specimens) of generalised melanism in Polynesian rats indicates such a recessive autosomal inheritance of melanism. The occurrence of two of these, plus the single piebald specimen, in the same archipelago might further indicate a general loss of heterozygosity in this remote destination.

In contrast, the absence of coat colour polymorphisms on Borneo and Sumatra may indicate a regular gene flow from the adjacent mainland populations in Southeast Asia. A similar lack of aberrant coat patterns in central east Polynesia (e.g., Society and Cook Islands, Samoa) might be explained by the existence of a broad interaction sphere or a large "homeland region", central to a number of smaller interaction spheres, as indicated by evidence from mtDNA phylogenies (*Matisoo-Smith et al., 1998*), with the concurrent regular gene flow. By contrast, the Polynesian rats of Hawaii have close affinities with those of the Marquesas and the Society Islands, but whether gene flow continued after the initial introduction(s) remains unclear (*Matisoo-Smith et al., 1998*). The occurrence of a piebald specimen favours the hypothesis of a high degree of isolation. The limitation of rats with white patches on their dorsal pelage to Luzon, Sulawesi and Java might confirm the existence of a regional interaction sphere including the Philippines, Borneo and Sulawesi, unrelated to that of Oceania, as inferred from the restriction of mtDNA haplogroup I to the Philippines, Borneo and Sulawesi (*Matisoo-Smith & Robins, 2004*); a cautionary note: Java

was not sampled in the latter study). Cytochrome B haplotype distributions also confirm a connection between the Philippines (Negros), Borneo and Sulawesi, all sharing H1, which occurs nowhere else (*Thomson et al., 2014*). Saipan (Mariana Islands) has the basic Pacific haplotype (H8), which gives no further clue on phylogeny; the other islands of my dataset were not sampled.

## CONCLUSIONS

Results revealed that populations of the Polynesian rat introduced onto Southeast Asian and Pacific islands show a trend towards a higher incidence of coat polymorphisms, such as white patches, piebald coats, leucism and melanism as compared to mainland populations from their native range, although not statistically significant. No correlation could be found between this incidence and time in isolation on the focal island. A negative correlation was found between the number of native predators and the incidence of aberrant coat patterns. The higher incidence of abnormal coat patterns on islands with a high level of native murid biodiversity is here tentatively explained as linked to a more exclusively human commensal life. The white patches, including piebald patterns, observed in Polynesian rats, are here explained as a pleiotropic effect of unintentional selection for tameness (commensal pathway of domestication), possibly accelerated by insularity (limited gene flow, lack of predators), whereas melanistic coat patterns are more likely the effect of isolation on gene frequencies and distribution (founding events) and thus not necessarily linked to increased tameness.

## ACKNOWLEDGEMENTS

I thank Darrin Lunde (United States National Museum of Natural History), Neil Duncan, Ross MacPhee, Marisa Surovy, Eleanor Hoeger, Brian O'Toole (American Museum of Natural History) and Colin Miskelly (Te Papa Tongarewa National Museum of New Zealand) for permission to access the collections under their care and their technical assistance. I thank Stergio Intzes for his assistance with the statistical analyses, George Lyras for making the figures and Tamatea McGlinn for proofreading the text. A special thanks goes to Bartholomeus van der Geer for much-appreciated technical support, making the photographs and discussing the initial idea. The reviewers are thanked for their thorough and critical reviews, which helped to significantly improve the manuscript.

### Funding

This work was supported by the Netherlands Organisation for Scientific Research (NWO grant 016.veni.181.041). The funders had no role in study design, data collection and analysis, decision to publish, or preparation of the manuscript.

### Grant Disclosures

The following grant information was disclosed by the author:
Netherlands Organisation for Scientific Research: 016.veni.181.041.

## Competing Interests

The author declares there are no competing interests.

## Author Contributions

- Alexandra A.E. van der Geer conceived and designed the experiments, performed the experiments, analyzed the data, contributed reagents/materials/analysis tools, prepared figures and/or tables, authored or reviewed drafts of the paper, approved the final draft.

## Data Availability

The raw data are available in the Supplemental Files.

## Supplemental Information

Supplemental information for this article can be found online at http://dx.doi.org/10.7717/peerj.6894#supplemental-information.

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
