# Peer review of "Effect of isolation on coat colour polymorphism of Polynesian rats in Island Southeast Asia and the Pacific"

_PeerJ, doi:10.7717/peerj.6894_

## Round 0.1 · original submission · Major Revisions

Please pay special attention to comments about confusion between the various drivers and syndromes and the necessary limits of the type of observational study your use to tease these different factors apart. Also, please note comments about the clarity of the writing and use of language.

Reviewer 1 ·

Basic reporting

I’m not a native speaker myself, but I had the impression that the text might profit from being proof read by a native speaker, as I think the use of words and grammar is incorrect in several places. Further, many sentences are incomplete and confusing.

The article includes sufficient introduction and background to demonstrate how the work fits into the broader field of knowledge. Relevant prior literature is appropriately referenced.

The structure of the article is conform to an acceptable format of ‘standard sections’.

The figures are relevant to the content of the article and of sufficient resolution. However, the Supplementary figures have no captions.

The submission is ‘self-contained,’ represents an appropriate ‘unit of publication’, and includes all results relevant to the hypothesis.

Experimental design

The research question is well defined, relevant & meaningful. It is not directly stated how research fills an identified knowledge gap, but this becomes evident from the introduction.

Rigorous investigation is performed to a high technical & ethical standard.

Methods are described with sufficient detail in most parts (others are specified in the general comments below) & information to replicate.

Validity of the findings

The data on which the conclusions are based is provided, as far as possible (no photographs from every single specimen are provided, but this is probably not feasible). The data is robust and statistically sound (as far as possible to tell, but I think more details should be provided, see below).

Conclusion are well stated, linked to original research question & limited to supporting results. However, I have several concerns regarding underlying concepts and limitations of the data, which are not sufficiently discussed (see general comments below).

Additional comments

The author investigated coat colour in a great number of Polynesian rat specimens from mainland areas and islands. The goal of the study was to investigate, if aberrant coat colour (e.g., white patches, lighter or darker coloration) occurs more often in the island populations than in the mainland ones, as a result of release from predator pressure and in analogy to domesticated animals. It was found, that aberrant coat colour occurs only on the islands, but hat this occurrence is not causally linked to island size, persistence time on the island, and number of predators and competitors. Instead, it is hypothesised that the observed changes are results of the proximity to humans and unintentional selection for tameness, limited gene flow, and genetic drift.

The research question is very interesting and worthwhile, as it uses the analogy of domestication and island evolution to elucidate fundamental connections between phenotypic traits and their mechanistic underpinnings. The sample is great, the data sampling seems sound, and the analyses appropriate. However, I think that shortcomings of the material should be discussed, as they are crucial for the validation of the results and the conclusions. In particular, the observed pattern of no aberrant colours in mainland specimens could be the result of a sampling bias (particular collectors prefer specimens with white patches over the “normal” ones) and/or a smaller sample size in the mainland group compared to the number of specimens from the islands would automatically increases the probability to find aberrant coat colours in island specimens. Such discussion of limitations are lacking.
Further, I feel that there is a bit of a confusion among concepts, particularly “commensalism”, “commensal pathway of domestication”, and “island evolution”. I am sure that the author knows the differences, but it is confused in the text. For example, commensal house mice (Mus musculus) on the continent are not considered to be selected for tameness along the commensal pathway of domestication. Instead, M. musculus is considered to be a commensal wild species (see e.g., the study on white patches in wild mice selected for tameness by Geiger et al. 2018). On the other hand, the author describes the Polynesian rats on the islands as human commensals that are therefore selected for tameness along the commensal pathway of domestication. This inconsistency between the concept of “commensalism” and “domestication along the commensal pathway” between these species should be discussed. (It might help to clarify things if you would further describe the nature of the commensal relationship between the Polynesian rats and humans. Are they living in a closer relationship than e.g., house mice and humans? Are there more/closer interactions? I am sure this is difficult, if not impossible to answer. But such issues should be mentioned and discussed to be able to fully evaluate the results). Second, I think that domestication and island evolution, although potentially resulting in similar phenotypic traits, are fundamentally different processes. On islands, there is normally no selection for tameness, in contrast to domestication. According to the neural crest cell hypothesis of the domestication syndrome (Wilkins et al. 2016), changes of neural crest cell migration would therefore not play a role in island evolution. Instead, and as also suggested by the author, non-adaptive factors (e.g., genetic drift) and predator release may play a role in generating the peculiar traits of mammals on islands. These differences should be more clearly described in the text. In the manuscript, it is stated at several places that the underlying mechanisms between the domestication and the island syndrome are the same (e.g., L. 18 – 19, 96 - 99).

Lastly and as stated above, many sentences are incomplete and confusing and should be edited.
Further, I’m not a native speaker myself, but I had the impression that the text might profit from being proof read by a native speaker, as I think the use of words and grammar is incorrect in several places.

These are my main concerns about the manuscript. Please find more detailed comments below:

L. 17 and throughout the text: “domestication syndrome” is sometimes accompanied by apostrophes and sometimes not. Please be consistent.
L. 18: “Simultaneous” implies a temporal component, which is not the case in this context. Maybe exchange with “concurring” or “co-occurring”?
L. 21: “their native range” instead of “its native range”
L. 22: It is not clear what you mean by “this”. Do you mean “aberrant coat colour has been found in …”? Specify please.
L. 24: I think that the convention is to write out numbers below 10. So this would mean “seven” instead of “7”. Further, do you mean “in the course of seven centuries since colonialization of these islands”? Please specify. Moreover, I think that this sentence is a bit misleading, as it implies that you were investigating rats from different time periods, dating back several centuries, which was not the case (?). Consider re-phrasing please.
L. 28 – 29: What is meant by “relaxed selection under commensality”? Less predation?
L. 49: include “and” before the last point of the list.
L. 52: in some cases there is evidence for an increased reproductive output. See e.g., Raia et al. 2003 (Evolutionary Ecology 17: 293–312)
L. 56: “melanisme” is without an “e” at the end
L. 61: Include a comma before “or”
L. 62 onwards: It might be interesting to mention e.g., the Falkland island fox as an island species that has previously been described as exhibiting domestication features (Clutton-Brock J, 1995, Science: 7-20)
L. 64: Replace square bracket in the citation.
L. 64: As far as I know, the term “domestication syndrome” has been used for the first time in animals by Wilkins et al. 2014 (Genetics, Vol. 197, 795–808) on the basis of a term which has originally been coined for crops. The cited works by Zeder do not use the term “domestication syndrome”.
L. 68: Why “amongst others”? As far as I know the only selection criterion was tameness/aggressiveness.
L. 77 – 78: This sentence is a bit weird and does not make much sense. It is particular changes of the neural crest cell migration, which might contribute to the characteristics of the domestication syndrome.
L. 83: The selected trait in this case is tameness. It might make it easier for readers to say so, instead of stating “traits that were directly or indirectly selected for”
L. 85: There is an “of” missing after “irrespective”.
L. 89 and 100: Please be consistent with brackets in the cases of the citations of the people that described the species.
L. 95: What do you mean by “spontaneous” evolutionary change in this context?
L. 96: Consider rephrasing the sentence to: “evolutionary changes in murids, which are typical for the domestication syndrome”.
L. 96 – 97: As already mentioned above, I don’t agree that the domestication syndrome and the island syndrome have a “partial developmental similarity”. The phenotypic results of domestication and island evolution are similar, as pointed out earlier during the introduction, but I don’t see why the developmental underpinnings should be the same. In the commensal pathway of domestication, there is selection for tameness and maybe alterations of the neural crest cell development. On islands, there is no selection for tameness and therefore probably not such a similar developmental link. I would therefore rephrase this sentence and not use the term “development”.
L. 101 – 104: This sentence is a bit confusing. Why should domestic rats develop depigmentation in parallel with other domesticates? After all they are domesticates already and you described earlier that depigmentation occurs in rats. I think this sentence is not needed at all.
L. 106 onwards: Is this species presently found on all these islands? Please specify the time period.
L. 109: do you mean it lives as “human” commensal? Please specify. Further, why do you list “scrubs” as an anthropogenic habitat? Could you please specify this further?
L. 111: What do you mean by “predator”? Are they regularly hunted by humans for food or as pests? Are they hunted with traps or otherwise?
L. 115 – 116: This sentence seems not complete: Compared to what? Further, you do not describe here why island area, which is analysed later on, is important in the context of the research question. Please add a background here.
L. 120-122: You refer to “specimen numbers” twice and to different Tables, Data S1 and Appendix S1? Which is the right one? Also, there is no file called “Appendix S1” or “Appendix S2” etc. Could you please provide unequivocal file names?
L. 124 and following: Would it be possible to indicate these island groups in Fig. 1 and its caption? It’s quite hard to follow you here.
L. 129 – 132: Could you provide references here, please?
L. 146: How did non-specialists evaluate this? Did you let non-specialists look over a certain portion of the furs? How do you define “significant” difference (usually, this term is only used for statistics)
L. 152: This white belly only occurring in the Flores population is very interesting in itself, in the context of the research question, isn’t it? It might be worthwhile to discuss this further.
L. 160 – 170: Could you please describe the statistical tests in more detail? E.g., how did you define the groupings for the tests and why did you use a Kruskal-Wallis Test for islands and a Mann-Whittney U test for island area?
L. 163: “evenly” instead of “evently”
L. 165 and otherwise: I’m not sure if I’ve seen this notation of a superscript “10” in log10 before. Isn’t it usually subscript or on the same level?
L. 173 -174: Do you mean Figure 2 instead of 3 here?
L. 174 – 176: I think I understand why you exclude the albino from the analysis (because albinism is not considered a part of the domestication and island syndromes). But please describe this further here or in the discussion, as it is quite important. Further, and as described below, I do not see the differences between the albino and the leucistic individual on the figures. Could you please specify?
L. 187: This sentence is incomplete.
L. 188: I do not understand this sentence. Do you mean that the difference of distribution of aberrant coat colour over the regions is not significant?
L. 201 and following discussion: There were many more specimens from islands than from the mainland. Could it be that there are more aberrant colour patterns on islands by chance? Please discuss why or why not.
L. 203 – 207: This sentence is somehow inconsistent: Do you argue that there is no correlation between aberrant coat colour and time in isolation because on the first and last colonized islands there was a similar amount of specimens with aberrant coat colour?
L. 208: Do you mean “predator” instead of “predature”? Where are these results?
L. 220 – 221: Could you please elaborate on why you think that this is an artefact? This is not clear from the text.
L. 225 – 226: It is not clear what you mean here. What do you mean by “insularity itself”? The influence of isolation on coat coloration via genetic drift and isolation? You might repeat here that all island rats are human commensals.
L. 239: Do you mean “particular” instead of “preferred” (which implies an intention)
Further: As far as I see, Geiger et al 2018 did not describe the nature of the colour polymorphism, which they observed in the mice. I mean, there is no quantification or description of the location of the white fur. They merely stated that there are white dots and patches of fur in some specimens. So it is not really clear if these patches resemble the ones in brown rats or the Polynesian rats and this comparison is therefore not feasible.
L. 244: For house mice, commensalism alone has not changed coat colour, but selection for tameness, i.e., a much closer relationship to humans than just the general proximity has probably resulted in coat colour changes (see Geiger et al. 2018). Do you think that this might be different in the Polynesian rats?
L. 245: The commensal pathway of what? Domestication?
L. 248 – 249: But you found that number of predators is not correlated with the occurrence of the aberrant coat colour. This sentence contradicts your results. Are there more predators on the mainland than on the island(s) with the highest number of predators? If yes, please present these numbers and discuss.
L. 255: Why is this (what exactly do you mean by “this”) not the case on Batam island? Because there are no predators anyway that would make such a coloration adaptive? Or because the vegetation/geology is different?
L. 262: What do you mean by “a higher degree of domestication”? I think these Polynesian rats are not domesticated, but commensal wild populations. Do you mean a greater number of characteristics that occur in the domestication and island syndromes?
L. 263: shift “observed” after “The” and before “colour polymorphism”.
L. 264: There is two times “distribution” in the list.
L. 303 – 304: Switch the positions of “further” and “might”
L. 313: delete “here”
L. 326: delete either “as” or “compared to”
L. 326 – 328: But you found a correlation between the colour polymorphism and island area and number of competitors. So this is contradicting. Do you mean a causal correlation?

Fig.1:
- Many geographic areas mentioned in the legend are not indicated on the map (e.g. Greater and lesser Sunda islands). It would help the reader a lot if you could indicate those on the map.
- “…Sulawesi, Moluccas etc.”: What do you mean by “etc.”? Specify please, or mention just the most important islands.
- Lines 4 and 5 in the caption: “after which” and “subsequently” mean the same and one of these expressions should be removed from the sentence.
- Consider splitting the second sentence of the caption in several sentences to make it easier to follow.

Fig. 2:
- What do you mean by “larger” geographic region? Consider just deleting “larger”
- Delete “Data for” in the caption title. It is not needed.
- There is something missing in the last bracket: “details of ventral (A) and dorsal skins (B-)”
- it is very hard to interpret the figure if it is in greyscale (printed).
- Would it be possible to provide a picture of normal coat colour here? This would make the comparisons more straightforward.

Fig. 3:
I do not see the differences between leucism and albinism that you describe in the materials and methods section. On what basis did you classify these specimens as albinistic/leucistic?

Citations:
The format is somewhat inconsistent. Publications with three authors are sometimes cited with all names in the text (i.e., x, y & z 2018) and sometimes abbreviating all names but the first (i.e., x et al. 2018). Please make this consistent according to the journal requirements.

Supplementary Figures:
- I did not find figure captions and could not follow the descriptions in the text. Could you please provide captions and/or indicate the patterns which the reader is supposed to see with arrows?
Further, it is quite hard to keep an overview over the many figures in the main text and the supplement. I would suggest to put also the supplementary figures in the main text and combine figures to make it easier for the reader to keep an overview. Raw data has been made available.

Reviewer 2 ·

Basic reporting

This interesting manuscript examines the distribution of aberrant fur color patterns in a widespread rodent across continental and insular settings. An impressive number of specimens was examined, with all relevant records of aberrant color patterns being from islands. I particularly like the link between the island syndrome and the domestication syndrome that the author makes. I have several specific comments, which the author could consider. See general comments for the author, below.

Experimental design

No experiment was conducted. The study simply compares continental and insular specimens of rodents, which is a valid and interesting scientific comparison in this manuscript.

Validity of the findings

I recommend analyzing the data with a different statistical method. See general comments for the author.

Additional comments

The manuscript is rather long, given the results. Some of the Introduction and Discussion could be pared down, particularly the rather lengthy discussion of the genetic basis of aberrant fur colors.

There is a potential collection bias in the specimens examined. I recall speaking with an old-style museum collector (long since deceased) of mammals, who said that they captured so many rodents at times that they would simply discard those that they considered to be of no interest. It is possible that such collectors fanning out across Southeast Asia and the Pacific islands would consider either wild type specimens or mutant specimens to be of no interest, thereby introducing a collection bias into the data base. I am not suggesting that it actually happened, but the possibility exists, and the audience perhaps should be aware of that.

The extremely small number of aberrant specimens would seem to render any statistical test just about unnecessary because of the low likelihood of finding any statistical significance. However, I suggest that the author could try a more powerful type of test, such as a linear or loglinear analysis of categorical data. Those tests also calculate a chi-square value, but they are based on maximum likelihood estimation and are typically considerably more powerful than a simple chi-square test, particularly if there are any cell sizes <5. There are two results where P<0.1, and a more powerful test may reveal significance, even with very small cell sizes.

On Line 99, the author states that a specific hypothesis was tested. It was tested only in a statistical sense because no experiment was conducted. I would use “address” or “examine” or some similar word rather than “test”.

There are a number of places where the grammar, spelling, and word usage could be improved. I will not edit the entire manuscript but will point out a few such places.

Line 53: “suit” should be “suite”.

Line 56: “melanisme” should be “melanism”.

Line 177: it is usually better to use “by contrast” rather than “on the other hand”.
Line 187 ends in mid-sentence, at least in my version of the manuscript.

Line 208: “predature” should be “predator”.

Line 228: something appears to be missing in that sentence.

Line 239: I would not use “preferred” in this context. “Specific” seems to be more appropriate.

Line 246: it is stated that the Polynesian rat is a human commensal where it has been introduced. I have captured many Rattus exulans on remote Pacific islands that are sparsely populated by humans, far from any human habitations or influences. While they may be frequent commensals of humans, they may often occur also in a completely sylvatic state.

Lines 297 and 299: I believe “mouse deer” should be “deer mouse”.

Line 322: “the other islands of our data set were not sampled.” I see only one author listed, so “our” should be “my”. The author should take full credit for assembling such an impressive data set.

---

## Round 0.2 · Minor Revisions

Please give special consideration to the comments of reviewer one and respond to their requests for new analysis. If you do not think these are warranted, please explain carefully. Otherwise, they made quite a few suggestions to continue to tighten the language. Please take a look at those and resubmit. Sorry for the delay in getting this back to you.

Reviewer 1 ·

Basic reporting

First of all, I would like to thank the author for considering all my suggestions for edits and answering my questions satisfactorily. The author has thoroughly revised the text and figures accordingly and I think that this very interesting and high quality piece of work has greatly improved, concerning content, style, and language. It is much more comprehensive now. However, a couple of new questions and suggestions arose, which I think would be important to consider as well (see below).

Please note that there are still no captions for the supplementary figures.

Experimental design

High quality, see first review

Validity of the findings

High quality, see first review. For exceptions, see general comments below.

Additional comments

My major comments/questions are the following:

(1) Why do you consider darker colour (melanistic variants) as an aberrant coat colour variation in the context of the ‘island syndrome’ in analogy to the ‘domestication syndrome’? Usually, these ‘syndromes’ are associated with lighter colours only, not darker ones. It would be important to elaborate on this.

(2) There is another important concept/cause for the observed pattern of aberrant coat colours in island populations, which is – as far as I could see – not discussed (only mentioned briefly in the abstract and aims): ‘predator release’, i.e., the lack of predator pressure and thus certain selection pressures. There are only few or no predators on most of these islands, as the author shows, and this might lead to a lack of selection against colour patterns that provide less effective camouflage, e.g., light fur, white spots. This, in turn, might lead to a relatively higher occurrence of these coat colour variants in the island populations, compared to the mainland populations, where such aberrant patterns might be selected against. This mechanism has been shown to be a probable cause of white or light colour variants in domestic mammals, including the house mice in the study by Geiger et al. 2018 (besides the selection for tameness).

For this, it would be important to repeat the analysis of a correlation between aberrant coat colour and number of predators including the number of predators on the mainland, which have not been considered for this analysis as far as I could gather from Table 2. (Please correct me if I am wrong.)

In this context – and also for other considerations – it would be important to discuss domestic predators (cats, dogs, ferrets etc.), which also inhabit anthropogenic environments, just as the rats. Are there domesticated predators on these islands? What might their influence be? The absence of all predators, including the domestic ones, is an important prerequisite for the selection for tameness.

(3) Although you did not find a significant higher occurrence of aberrant coat colour patterns on islands compared to the mainland, you still state in the abstract, discussion, and conclusions, that there is a higher occurrence of such aberrant coat colours on the islands. This is inconsistent and a bit misleading. Could you please include qualifiers such as e.g., “there is a trend towards a higher incidence of aberrant coat colours on the islands compared to the mainland, although not statistically significant”?

(4) The discussion is still a bit convoluted and therefore hard to grasp. I would therefore suggest summarising certain parts, e.g. the paragraphs about competition and predation could be summarised and discussed together and the genetics part is a bit lengthy and might profit from summarising.


Here are some more minor issues:

- For enumerations, you sometimes use a comma before and/or, and sometimes not. Please be consistent.

- L. 92: The citation by Lomolino needs a page number.

- L. 93: “wolf” should be in lower case.

- L. 94: Falkland wolves also had white on the muzzle and on the lower limbs, not just the tail tip (see Clutton-Brock 1977, Man-made dogs)

- L. 169 ff.: You’ve explained to me that on some islands, lighter specimens are the norm, on others darker ones are the norm: “The ‘normal’ coat colours differ substantially between the regions,... For example, New Zealand specimens are much darker, towards melanism, yet are considered ‘normal’, whereas exactly such a specimen from an area with only light coloured furs would be labelled as melanistic.” Would this mean that the “very dark, toward melanistic” specimens from Polynesia in Table 2 would be considered normal in New Zealand. If this is correct, it would be important to mention here what you explained to me.

Further, although you deleted this part in the newest version, I think it would be an important to mention that you also showed specimens to curators and co-workers. This qualifies your own evaluation. Therefore, I would just write here what you explained to me.

- L. 177 – 188 and 195 - 204: These parts belong to the discussion.

- L. 180 and elsewhere in the text (probably as the result of “track change” in word): Missing space after “.

- L. 215: number of aberrant coat patterns “per biographic area”? Please specify.

- L. 223 – 224: I would exclude the statement about summary statistics in Table S1 from this bracket, as it also concerns information about results not related to the geographic distribution.

- L. 226 & 244: coat “colour” pattern?

- L. 227: Delete “further”. At this point in the manuscript you have technically already done all the analyses.

- L. 233: You could add a reference to Fig. 1 where you mention “a few strands of hair” to illustrate what you mean.

- L. 235: You could add a reference to Fig. 4 here, for comparison.

- L. 236 – 239: You explain that albinism and leucism are two completely different phenomena (to which I agree). So why do you explain leucism on the basis of albinism here?

- L. 238: Which “evenly buff coloured specimens” do you mean here? From which area are they? Please specify.

- L. 239 ff. These specimens lack eyes. So why do you know that they “have black eyes”?

- L. 252: log transformed “island” area?

- L. 267 – 269: This sentence is a bit long and confusing. Consider splitting.

- L. 274 - 278: You mix predation and competition in this sentence. Which one do you mean?

- L. 278 ff.: You are quite negative here, although there is apparently plenty of evidence that these rats are more associated to humans in the introduced areas, due to competition with native rodents. Similar scenarios have also been suggested for commensal house mice in Europe, which are usually outcompeted by other rodent species in non-anthropogenic habitats (e.g., Berry RJ et al. 1982. Colonization by house mice: an experiment. Journal of Zoology 198.3: 329-336.)

- L. 291 – 293: You could refer to Figures 1 and 2 here, for comparison.

- L. 306 – 308: Only the piebald specimens from Hawaii show these coat colour patterns “resembling colour polymorphisms of R. norvegicus”? Please specify what you want to express with this sentence.

- L. 308 ff.: White patches are not only a result of the commensal pathway of domestication, but for domestication in general, i.e., selection for tameness (e.g., domesticated foxes exhibit white spots, although they have been domesticated along the “directed pathway” according to Zeder).

- L. 323 – 327: This part contradicts what you say in the last sentence of the previous paragraph. If there are no predators on Hawaii, why should there be an adaptive advantage of having the same coat colour as the rocks?

- L. 388: “The occurrence of a piebald specimen favours a high degree of isolation.” does not make much sense. Do you mean: The occurrence of a piebald specimen favours “the hypothesis” of a high degree of isolation”?

Conclusions: As described above, this summary does not correspond to your results. According to your statistical tests, there was no higher incidence of coat polymorphisms on the islands. Additionally, you did indeed find a correlation of colour polymorphisms and number of competitors, although you describe here that there was no correlation. Please clarify.

Figures 3 – 5: Not sure if this is necessary, but it might be useful for readers if you shortly note that all these genus and species names on the specimen labels on the photographs are synonymous to R. exulans according to Wilson and Reeder 2005 (?)

Supplementary Data S1 has no captions on top of the columns. Please add those.

Supplementary Table S2: The last header “Relation between island area and aberrant coats” should probably just be “Relation between island area” (because you also test for relations between island area and number of competitors)

Reviewer 2 ·

Basic reporting

Refer to comments in my original review.

Experimental design

Refer to comments in my original review.

Validity of the findings

Refer to comments in my original review.

Additional comments

I am largely satisfied with the revisions made by the author, and I found this study to be most interesting. I applaud the author for addressing an interesting topic, compiling an impressive data set, and appropriately revising the original manuscript. I recommend making a few mostly editorial changes.

Line 57: “and dispersal abilities” should be changed to “and reduced dispersal abilities”.
Line 70: “reduction of brain size” should be changed to “reduced brain size”.
Line 71: “change “depigmentation of” to “depigmented”.
Line 98: I think “dog” should follow “feral domestic”.
Beginning of Materials & Methods: collections housed in two museums are mentioned here, but in the new paragraph from lines 210-219, the author points out that there are three collections. In the Acknowledgements, the National Museum of New Zealand is mentioned, so assume that it is the third collection, and it should be added to the beginning of the Materials & Methods. Also, if the new paragraph is correct in stating that there are three collections, then “between” on lines 218 and 219 should be “among”.
Line 150: “Smithsonian Institution” rather than “Smithsonian Institute”.
Line 155: “are as follows per group:” (add the colon :).
Line 169: “faunas” rather than “fauna”.
Lines 228-233: I am now confused about the wording of the statistical tests. An association is a correlation, but a Kruskal-Wallis test is a nonparametric test for differences among medians of more than two samples (analogous to a parametric ANOVA for testing for differences among means). A Mann-Whitney test does not test for a correlation but instead is a nonparametric test for a difference between medians of two samples (analogous to a parametric t-test). I believe some clarification or different wording is in order here. The same comment applies to the results section (lines 265-271) and lines 282-283 in the Discussion. Also, I believe that the test statistics that are associated with the P values should be given in lines 265-271.
Line 279: “amount” should be “number”.
Line 285: Change “This correlation is best explained as coincidental, not causal.” To “This correlation is best explained as coincidental rather than causal.” Also, verify that “correlation” is the correct word here.
Line 306: a comma (,) is needed after “Luzon”.
Line 372: a period (.) is needed after “(Wagner, 1845)”
Line 399: change “population” to “populations”.
Line 439: change “(founding events), thus” to "(founding events) and thus”.

---

## Round 0.3 · accepted · Accept

I apologize again for the delay in getting the last round of comments to you - and I appreciate how carefully you attended to the additional requestions.

#